# Research on Atomic Oxygen Erosion Influence of Structural Damage and Tribological Properties of Mo/MoS_2_-Pb-PbS Thin Film

**DOI:** 10.3390/ma15051851

**Published:** 2022-03-01

**Authors:** Cuihong Han, Guolu Li, Guozheng Ma, Jiadong Shi, Aobo Wei, Zhen Li, Qingsonge Yong, Haidou Wang, Huipeng Wang

**Affiliations:** 1School of Materials Science and Engineering, Hebei University of Technology, Tianjin 300401, China; hanmutou@163.com (C.H.); jiadong1207@126.com (J.S.); weiaobo0625@163.com (A.W.); 2School of Mechanical Engineering, Tianjin University of Technology and Education, Tianjin 300222, China; 3National Key Laboratory for Remanufacturing, Army Academy of Armored Forces, Beijing 100072, China; wanghaidou@tsinghua.org.cn; 4State Key Laboratory of Mechanical Systems and Vibrations, Shanghai Jiao Tong University, Shanghai 200240, China; lizhen2019@sjtu.edu.cn; 5China Aerodynamics Research and Development Center, Mianyang 621000, China; qing-song_yong@163.com; 6National Engineering Research Center for Remanufacturing, Army Academy of Armored Forces, Beijing 100072, China; 7School of Mechanical and Electrical Engineering, Jiangxi University of Science and Technology, Ganzhou 341000, China; wanghuipeng1983@126.com

**Keywords:** atomic oxygen, Mo/MoS_2_-Pb-PbS composite film, oxide, lubricating components

## Abstract

To investigate atomic oxygen effects on tribological properties of Mo/MoS_2_-Pb-PbS film and further enlarge application range, atomic oxygen exposure tests were carried out for 5 h, 10 h, 15 h, and 20 h by the atomic oxygen simulator with atomic oxygen flux of 2.5 × 10^15^ atoms/cm^2^·s. The exposure time in test was equivalent to the atomic oxygen cumulative flux for 159.25 h, 318.5 h, 477.75 h, and 637 h at the height of 400 km in space. Then, the vacuum friction test of Mo/MoS_2_-Pb-PbS thin film was performed under the 6 N load and 100 r/min. By SEM, TEM, and XPS analysis of the surface of the film after atomic oxygen erosion, it was observed that atomic oxygen could cause serious oxidation on the surface of Mo/MoS_2_-Pb-PbS film, and the contents of MoS_2_, PbS, and Pb, which were lubricating components, were significantly reduced, and oxides were generated. From AES analysis and the variation in the main element content, Mo/MoS_2_-Pb-PbS thin film showed self-protection ability in an atomic oxygen environment. Hard oxide generated after atomic oxygen erosion such as MoO_3_ and Pb_3_O_4_ could cause the friction coefficient slight fluctuations, but the average friction coefficient was in a stable state.

## 1. Introduction

Friction parts are the important component of space equipment such as manipulator arms, solar arrays, satellite attitude adjustment mechanisms, and driving mechanisms for space exploration instruments and communication antennas, etc. [1,2]. The space environment factors such as atomic oxygen (the abbreviation is AO), vacuum, and irradiation directly affect the friction lubrication performance. At the same time, the space friction device has almost no maintainability in the service cycle; hence, the space lubrication materials put forward higher requirements. When spacecrafts are at the speed of about 8 km/s in low earth orbit (LEO), the AO density on the windward surface of the spacecraft will increase to the order of 10^12^–10^15^ atoms/cm^2^·s, and the collision kinetic energy is as high as 5 eV, which is equivalent to the high temperature effect of 4.8 × 10^4^ K [3,4]. This energy value is high enough to break the chemical bonds of most materials commonly used in space applications. Therefore, AO is considered as one of the most dangerous environmental factors in LEO. Further research on the damage mechanism of material tribological properties under AO can ensure the reliability of space equipment [5,6].

MoS_2_ is widely used as lubricating film in space components because of its ultralow friction in ultrahigh vacuum and inert gas environment [7,8,9]. However, MoS_2_ film has a high density of pores structures along the boundary of columnar grain, which provides a reaction highway to the erosion process; MoS_2_ film can be oxidized easily by AO so that lubrication properties of MoS_2_ film is weakened. During space service, components are inevitably bombarded by AO, which requires the coating to have high oxidation resistance. To improve oxidation resistance and tribological properties of MoS_2_ film, researchers have explored metal and ceramic incorporated as well as multilayer structures by element doping modification such as Ti, Cr, Au, Al, and Nb [10,11,12]. There were no obvious changes in the morphology, phase structure, element composition, and friction property observed from the space exposed and non-exposed composite films. So, it indicated the composite film can exhibit better anti-oxidation ability when exposed in an LEO environment. Researchers have investigated the structure and property changes in MoS_2_ film exposed to AO flux, which has a similar density and energy to that of the low earth orbit. The pure molybdenum disulfide film system has been studied by Wang P, the result showed the diffusion of AO can reach 600 nm, which is much higher than the depth of AO at energy (5 eV), and is also higher than the 2–5 nm considered in the common literature [13,14]. Meanwhile, they found that revising the MoS_2_ lubricant film by doping Ti atoms or especially fabricating in an MoS_2_/Ti multilayer structure can effectively improve the film resistance to oxidation in AO exposure [15]. Liam S. Morrissey [16] simulated the impact of AO on silver and aluminum using the Reax FF force field in molecular dynamics. The results showed that although erosion is an important parameter to measure the damage of materials by high energy impact, it is not enough to describe the damage amount and state of the remaining substrate. HaiFu Jiang [17] researched reduced graphene oxide paper (rGOP)’s surface structure and resistance characteristics by AO (AO) in a ground-based simulation device. They found the resistance data showed that R0/R has a linear relationship with the AO flux density. The maximum AO detect fluence reached 5 × 10^19^ atom/cm^3^ when the rGOP thickness was 0.8 μm, which implies that greater thickness is expected to improve space service life of rGOP. In most of the research, a ground-based AO (AO) simulation facility was used to erode lubricated film, molecular dynamics was also a helpful tool to simulate the AO erosion process. In order to improve oxidation resistance and good lubricating in air, Ren [18] deposited MoS_2_/Pb–Ti composite and multilayer coatings by unbalanced magnetron sputtering system. Furthermore, it is rarely mentioned that different AO erosion times will have an influence on the tribological properties of MoS_2_ film. In addition, the influence of Pb as doped elements on the nanocomposite and multilayer structure of MoS_2_ are still unknown.

In this paper, Mo/MoS_2_-Pb-PbS composite film was obtained by combining RF magnetron sputtering technology with low-temperature ion sulfide. The influence of AO exposure on the Mo/MoS_2_-Pb-PbS composite film’s compositional and structural changes was studied. The samples with Mo/MoS_2_-Pb-PbS composite film were exposed under a laboratory AO beam with a flux of 2.5 × 10^15^ atoms/cm^2^·s for setting time. The compositional changes due to AO exposure were evaluated by X-ray diffractometer (XRD) and energy dispersive X-ray spectrometry (EDS), and the structural evolution was investigated using scanning electron microscopy. The tribological properties of introducing Pb element and multilayer structure on the erosion behavior in a vacuum was investigated using MSTS-1 multifunctional vacuum friction tester. The changes in microstructure and friction properties of the film after AO erosion were studied, and then the damage mechanism of AO erosion was analyzed.

## 2. Experimental Details

### 2.1. Materials

In this study, Mo/MoS_2_-Pb-PbS composite film is combined on 9Cr18 (AISI440C) substrate by composite process of “PVD coating + low-temperature ion sulfide”, which is the multi-element composite solid lubricating film with double-layer superimposed structure. RF magnetron sputtering is firstly used to deposit Mo film (about 100 nm). Then Pb film is deposited on the Mo film. The papered Mo-Pb film (about 1900 nm) are treated by low temperature ion sulfurizing in order to obtain Mo/MoS_2_-Pb-PbS composite film, which are two-layer stacking structure. The single metal (Mo) bonding layer ensures strong bearing capacity, as well as high bond strength between film and substrate. Lubricating surface layer with Mo/MoS_2_ is the main body as well as dispersing Pb (PbS) particles, so Mo/MoS_2_-Pb-PbS composite film possesses low shear strength and good plastic deformation capacity. The process and parameters of RF magnetron sputtering and low temperature ion sulfurizing are mentioned in [19].

Compared by the tribological performance under dry air and vacuum environment (4 × 10^−4^ Pa) with MSTS-1 multifunctional friction tester [20,21,22], Mo/MoS_2_-Pb-PbS composite film shows good friction-reduction and anti-wear properties in vacuum (4 × 10^−4^ Pa). Therefore, the tribological performance of Mo/MoS_2_-Pb-PbS composite film under AO erosion is investigated in order to verify its workability in space.

### 2.2. Vacuum AO Simulation Device and Surface Analysis Methods

In order to research space friction performance of Mo/MoS_2_-Pb-PbS composite film especially under the AO (AO) erosion, the designed AO source is optimized and integrated with the MSTS-1 multi-functional vacuum friction and wear tester previously developed by our research group. The schematic diagram of MSTS-1 can be found in the literature [19]. The main structure of the AO source is composed of a microwave source, circulator, water loading, three screw regulator, plasma generator cavity, ECR magnetic field system, neutralization system, and pumping system, as shown in Figure 1.

The incident AO flux according to the above calibration procedure was 2.5 × 10^15^ atoms/cm^2^·s. During the calibration test, the irradiation distance was 190 mm, and the microwave output power was set to 300 W with a constant O_2_ gas pressure prior to plasma ignition of 0.05 Pa (gas flow 30 sccm). The uniform irradiation area of AO beam is more than 30 mm × 30 mm, and the average energy of AO is about 5–10 eV under the given experimental parameters.

Nano-SEM 450 field emission scanning electron microscope (with full quantitative energy spectrometer EDS, FEI Company, Hillsboro, OR, USA) was used to observe the microstructure and phases of the film surface, wear tracks, and counter-ground surface ingredients.

Rigaku D/MAX 2400 X-ray diffractometer (XRD) (Rigaku, Tokyo, Japan) was mainly used for phase qualitative and quantitative analysis, which was with Cu target, scanning speed 8°/min and step length 0.02°, under 40 kV, 40 mA, incident wavelength λ = 0.15406 nm.

Relative content and chemical valence of main elements on film surface was measured by ESCALab220i-XL X-ray photoelectron spectrometer (XPS) (VG Scientific, Waltham, MA, USA) with Mg-Kα excitation source, power about 300 W. The base pressure was about 3 × 10^−9^ mbar. The binding energies were referenced to the C1s line at 284.8 eV from adventitious carbon.

JEM-2100F high-resolution transmission electron microscopy (HRTEM) (Japan Electronic Corporation, Beijing, China) was used to obtain microstructure of composite films under acceleration voltage 200 kV, line resolution 0.1 nm, point resolution 0.23 nm.

PHI-700 nano scan Auger Electron Spectrometer (AES) (ULVAC-PHI, Kanagawa, Japan) was used to detect the distribution of elements in the film along the depth direction, using coaxial electron gun and CMA energy analyzer, scanning Ar^+^ gun high voltage 5 kV, and standard sample is SiO_2_/Si thermal oxidation.

### 2.3. Process

The four samples were exposed by the AO simulation device for 5, 10, 15 and 20 h in order, and the area of AO beam with AO flux value of 2.5 × 10^15^ atoms/cm^2^·s was a circular area with a diameter of 30 mm, and the area is about 7.065 cm^2^. The exposure time was equivalent to the AO cumulative flux of exposure for 159.25, 318.5, 477.75 and 637 h at the height of 400 km in space, where AO beam with a flux is 7.85 × 10^13^ atoms/cm^2^·s under moderate solar activity.

The tribological performances of the Mo/MoS_2_-Pb-PbS composite film after AO erosion were evaluated by MSTS-1 multifunctional vacuum friction tester (MSTS-1, Beijing, China) under vacuum environment (5 × 10^−3^ Pa). During the test, the load was 6 N (Hertz contact stress was 0.5366 GPa), the speed was 100 rpm, and the friction time was 1200 s. If the friction coefficient continued to exceed 0.6 in the test process, the test was stopped. Otherwise, the sliding friction was continuously carried out for 20 min. The upper sample of MSTS-1 multifunctional vacuum friction tester was a 9Cr18 bearing steel ball with the dimension of Φ9.525 mm, the hardness is HRC58, and the surface roughness is Ra0.025μm. The lower sample disc (Φ34 mm × 6 mm) was the Mo/MoS_2_-Pb-PbS multi-component composite film.

In order to investigate effect of AO erosion on structural damage and tribological properties of Mo/MoS_2_-Pb-PbS thin film, we chose the sample that was exposed for 15 h with AO as the research subject in following discussion.

## 3. Results and Discussion

### 3.1. Composition and Structure

After RF magnetron sputtering and low temperature ion sulfurizing, Mo/MoS_2_-Pb-PbS multilayer thin film was exposed to the AO with different incident fluence, and the corresponding composition and structure changes were investigated by SEM and EDS. Figure 2 shows the surface morphology and composition of Mo/MoS_2_-Pb-PbS film before and after AO exposure. The microstructure of the film without AO erosion was composed of nano-scale irregular particles with some pores between particles, as shown by Figure 2a. After AO erosion for 15 h, about 1.35 × 10^20^ atom/cm^2^ incident fluence exposure, the surface morphology Mo/MoS_2_-Pb-PbS film is shown by Figure 2d. The composition Mo/MoS_2_-Pb-PbS film without AO erosion is shown in Figure 2b. The contents of Mo and Pb were 41.29% and 38.26%, respectively. After AO erosion, the number of cauliflower-like large particles on the surface of the film decreased because of flash etching of the AO beam. Therefore, the film surface became flat and full of pitting. In order to study the composition changes in Mo/MoS_2_-Pb-PbS film further, XPS was used to RBS. Figure 2c shows the XPS results of the valence of the Mo and Pb elements on the surface of Mo/MoS_2_-Pb-PbS composite film before AO erosion. As shown in Figure 2c, the main peaks of Mo element on the surface of the film are mostly located at 230 eV, corresponding to Mo, the molybdenum compound of MoS_2_ and MoO_3_, respectively. Due to the inevitable contact with air during the preparation of XPS samples, the Mo element on the surface of the original sample was also partially oxidized to MoO_3_, but the high content of MoS_2_ is still the main component. After AO erosion, the contents of Mo and Pb were decreased to 18.73% and 18.33%, respectively, while the content of O was increased to 47.81%, because of the oxidizing reaction shown by Figure 2e. Combining with Figure 2f, the content of MoS_2_ decreased significantly because a large amount of Mo was oxidized to MoO_3_ and metastable MoO_2_ as shown in Figure 2f. An obvious feature in the spectrum is the increase in the content of elemental Mo on the film surface caused by AO erosion because of the sputtering etching effect of high-energy AO removed the S-rich layer on the film surface. The Pb element on the original film surface mainly exists in the form of elemental Pb, PbS, and PbO, and the content of PbO was small. After AO erosion, the Pb element on the film surface was oxidized to Pb_3_O_4_ and minor PbSO_3_, and the spectral peak of elemental Pb disappeared.

XPS results reveal clearly that mostly Mo and Pb elements in the film surface were both oxidized to MoO_3_ and Pb_3_O_4_, respectively. There were rare amounts of the simple substance Pb, and the content of MoS_2_ was decreased compared by Pb and Mo elements atomic percent form Figure 2b,e. The Pb decreased to 18.33% from 38.26 without AO erosion, and Mo element decreased to 18.73% from 41.29 without AO erosion. In order to get more information about the structural evolution, Mo/MoS_2_-Pb-PbS multilayer film before and after the AO exposure were investigated by TEM. Figure 3a shows that there were long-range ordered structures and glassy substances with different orientations and plane spacings in the Mo/MoS_2_-Pb-PbS film before AO erosion, and there were obvious interfaces between different structures. Glassy substances were one crystal structure of 2H, which is the stable state structure. Molybdenum disulfide crystal belonged to hexagonal system, with three crystal structures of 1T, 2H, and 3R. 1T-MoS_2_ and 3R-MoS_2_ crystal belonged to a metastable structure. After AO erosion, the original large-scale regular structure disappeared and the cluster size became smaller, while the glassy region became larger (the yellow circle region shown in Figure 3), and a small-scale ordered structure region appeared (the green circle region shown in Figure 3b), which is consistent with the test results of the decrease in metal (Mo, Pb) and metal sulfide (MoS_2_, PbS) in Figure 2, and the new generation of a large number of oxides (MoO_3_, Pb_3_O_4_, etc.). Highly active atomic oxygen impacts caused great heat, whereby 1T-MoS_2_ and 3R-MoS_2_ of metastable states transformed into stable 2H-MoS_2_. Hence, the glassy region became larger.

In summary, AO with high activity and high energy will cause serious oxidation on the surface of Mo/MoS_2_-Pb-PbS film, and the contents of lubricating components MoS_2_, PbS, and Pb were significantly reduced, and a large number of oxides was generated.

The element distribution along the depth direction of Mo/MoS_2_-Pb-PbS composite film after 15 h AO erosion was shown in Figure 4 by used AES analysis, when the Ar^+^ sputtering rate is 30 nm/min. The contents of Mo, Pb, and S elements in the top layer of the film are 18.1, 14.8 and 10.3%, respectively, and the content of O element is as high as 54.4%. With the sputtering etching depth increasing, the content of the Mo, Pb, and S elements increased gradually. However, the content of the O element decreased sharply. When the etching time was 3 min (about 100 nm from the surface), the contents of the S and Pb elements reached their maximum values 18.52% and 19.72%, respectively. Subsequently, the content of the S element decreased slowly, and the content of the Pb element was stable at about 18.5%. After sputtering for 10 min (about 300 nm from the surface), the content of each element tended to be stable, and the content of the Mo, S, and O elements was stable at about 65, 9 and 5.5%, respectively. In the range of 60–70 min, the contents of the Mo, Pb, and S elements were mutated. In this region, the contents of Fe and Cr, which are the main elements constituting the 9Cr18 matrix, were rapidly increased to about 70% and 10%, respectively. There were 1800–2100 nm away from the surface of the sample, which was close to the Mo/MoS_2_-Pb-PbS film thickness of 2000 nm. The position where the element mutation occurred was the film–substrate interface.

Previous research results showed that the S element content of ion-sulfurized film often had a decreased gradient in the depth direction, and the S element enrichment layer generally appeared near the surface in general [23]. As shown in Figure 4, the S element content of the film surface in the range of 300 nm was indeed relatively high, but the maximum S element content did not appear on the top surface. Because the standard formation free energy of oxides was low, it is very easy to form oxide. When the AO beam with high energy is on the surface of Mo/MoS_2_-Pb-PbS film, AO will “grab” some metal ions from the metal sulfides. Then MoO_3_ and Pb_3_O_4_ will be formed after AO enters the surface layer of the film. Therefore, it is obvious that the content of Mo, Pb, and S in the surface layer of the film will be low, and the content of O is high. However, with the increase in film depth, the oxidation decreased rapidly [24,25]^.^ Combined with the variation in the main element content, the Mo/MoS_2_-Pb-PbS film has a certain self-protection ability in the AO environment. Oxides such as MoO_3_ and Pb_3_O_4_ can hinder the deep oxidation of the lubricating material in the subsurface. The thickness of the oxidized area is clearly about 300 nm.

### 3.2. Tribological Properties

Because the remarkable tribological properties of Mo/MoS_2_-Pb-PbS composite film in a vacuum, the effect of the space environment and especially AO erosion through the top surface to depth in determining the film tribological properties are also discussed. Tribological tests were performed on a ball-on-disk tribometer in the low-pressure vacuum of 5 × 10^−3^ Pa, and the results are shown in Figure 5. The variation in the friction coefficient of Mo/MoS_2_-Pb-PbS film without AO erosion showed obvious three-stage characteristics of ‘starting-running-stability’. The starting friction coefficient was about 0.075, which rose rapidly to 0.175 when friction time was 30 s. Then, the friction coefficient decreased rapidly and stabilized to 0.05 when friction time was 80 s. After which, the fluctuation of the friction coefficient was very small, and the friction coefficient curve was very stable as shown in Figure 5c. We found that the starting friction coefficient of Mo/MoS_2_-Pb-PbS film after AO erosion increased to 0.2, and then the friction coefficient oscillated violently and decreased gradually. When the friction test time was about 330 s, the friction coefficient was stable at about 0.06. During the whole test process, the friction coefficient curve fluctuated slightly compared with the film before AO erosion.

Before AO erosion, obvious material transfer was observed on the worn surface of the film, which was seen as separate deep scars along the sliding direction. A large amount of material piled up outside the wear scar. In the vacuum environment, because the ambient medium was rarefied and there was no convection heat dissipation in air, the friction interface temperature was higher. The surface of the film was further softened during the friction process. The friction dual ball was embedded in the soft film surface under the positive pressure, and the soft film was pushed in the sliding to make plastic flow, and furrows were formed; meanwhile, a large amount of material accumulation was formed on both sides of the wear scar. Hence, the wear of Mo/MoS_2_-Pb-PbS film without AO erosion was dominated by plastic deformation and material transfer, and there were few lubricants separated from the friction orbit in the form of debris.

After AO erosion, the wear of the film became uneven, and the local material removal was serious. A plentiful supply of parallel grooves with different depths were distributed along the sliding direction. Irregular black particles could also be observed on the wear track. Compared with the three-dimensional morphology and profile data of the wear scars before and after AO erosion, the depth, width, and volume of the wear scars of the film after AO erosion were significantly increased, especially the depth of the wear scars, which increased from 0.618 to 1.287 µm.

Sputtering and oxidation of spacecraft surface materials by AO with high speed and high activity often lead to significant changes in the quality of materials [26,27]. As shown in Figure 6a, the quality of Mo/MoS_2_-Pb-PbS thin film increased after AO erosion at different times, because the molecular weight of the newly formed oxides, namely, MoO_3_ and Pb_3_O_4_, was larger than that Mo and Pb elements’ sulfides, and some elemental Mo and Pb were also oxidized. After AO erosion for 5 h, the sample mass increased by 0.6 mg. When the erosion time was extended to 10 h, the mass of the sample increased to 1.63 mg. When the erosion time continued to extend, the increase rate of sample mass decreased and stabilized at 1.7–1.8 mg.

As shown in Figure 6b, after AO erosion for different time, the average friction coefficient of Mo/MoS_2_-Pb-PbS film in the stable wear period remained around 0.06. When the erosion time was less than 15 h, the wear scar depth of the film increased approximately linearly with the extension of erosion time, but when the erosion time continued to increase (after 20 h), the wear scar depth no longer increased significantly.

### 3.3. Discussion

The microstructure of Mo/MoS_2_-Pb-PbS composite film changed significantly after AO erosion at different times, and the tribological properties were also degraded to varying degrees. In this section, we briefly analyze the mechanism of spatial AO on the structural change and tribological performance degradation of Mo/MoS_2_–Pb-PbS thin film.

Firstly, the preparation process and composition compatibility of Mo_/_MoS_2_-Pb-PbS thin film determine that the surface roughness of the thin film is relatively large, and there is a certain gap between the structural unit particles. Mo/MoS_2_-Pb-PbS thin film were prepared by the two-step composite process of ‘RF magnetron sputtering and low temperature ion sulfurization’. The Mo-Pb thin film was formed of atomic groups of Mo and Pb elements, which were sputtered from the target, in the way of layer-by-layer stacking and mutual doping. After sulfurization treatment, Mo and Pb in the Mo-Pb thin film were partially sulfurized; thus, the special structure was formed. The two-phase of Mo and Pb mixed structure was the bulk of this special structure, and rich in MoS_2_ and PbS metal sulfide as lubricating phases. The microstructure of Mo/MoS_2_-Pb-PbS thin film was metal and metal compound particles with nano-sized, and their agglomerated micron-sized particles. However, the important difference from the sputtering deposition of pure MoS_2_ thin film was that the Mo/MoS_2_-Pb-PbS thin film lacked coarse columnar crystal growth, and that there were no structural defects, such as penetrating deep holes, on the surface of the thin film.

Secondly, AO with high energy and high activity will sputter and etch the film surface; meanwhile, AO will react with elements on the film surface when the film is irradiated by AO. The components in Mo/MoS_2_-Pb-PbS thin film are metals or metal compounds, and each component is bound by strong chemical bonds. The atomic mass of Mo and Pb is large, and the kinetic energy is transferred to the surface of the thin film when the AO hits the surface of the film with high speed. These energies are not enough to destroy the chemical bonds inside the film, but the long-term physical sputtering will produce an etching effect on the surface of the film, making the film surface smoother and denser. After kinetic energy transfer, a lot of oxygen atoms are adsorbed on the surface of the film and the film surface is oxidized, as shown in Figure 7.

When AO reached the film surface, it preferentially reacted with particles on the top of surface, so severe oxidation occurred at the top of the film. With the increased AO cumulative flux, some of the oxygen atoms will fill the gap of particles on the surface of the film, and even migrate to the film interior along the shallow defects of the film until the reaction with the film stops moving. The filling and migrating of AO also oxidizes the film interior. After AO reacts with Mo, Pb, MoS_2_, and PbS in the film, a thick and tightly bonded oxide layer is formed on surface of the film, and the quality of the film is significantly increased.

Thirdly, according to the molecular random dynamics model and chemical reaction dynamics model’ of the interaction between AO and the surface of the film, the oxidation degree of the film in the AO environment was closely related to the diffusion force of AO into the film and the activation energy of the oxidation reaction. The oxidation ability of AO in the thermal state was strong. With the increase in erosion, the continuous coverage of oxide thin layers such as MoO_3_ and Pb_3_O_4_ will be formed on the surface of the film. These formed dense oxide layers can prevent the diffusion of oxygen atoms into particles. There was also no penetrating deep hole in the film as a ‘channel’ for oxygen atoms to diffuse deeply. However, the Mo/MoS_2_-Pb-PbS thin film will undergo seriously oxidation under the action of AO, even if the erosion that increased the oxidation damage of the film was limited to hundreds of nanometers on the surface of the film.

After AO action, the starting friction coefficient of the film Mo/MoS_2_-Pb-PbS film increased due to the hard oxide shell formed on the surface of the film. As the sliding friction progresses, these oxide thin layers were broken and removed quickly. The hard oxide particles embedded in the surface of the soft lubrication film, which increases the fluctuation of the friction coefficient; however, the average friction coefficient will not change significantly in the stable period.

## 4. Conclusions

The main conclusions are as follows:The quality of the Mo/MoS_2_-Pb-PbS thin film increased. Partial oxidation occurred because of AO erosion, and dense oxide film, such as MoO_3_ and Pb_3_O_4_, was formed.The starting friction coefficient of the film increased after AO erosion because the lubricating of these oxides was weaker, but with the removal of the oxide thin layer, the friction-reducing lubrication performance of the film recovered quickly.Oxides can also prevent the metal and elements inside the film structure from being oxidized by AO, and the oxidative damage is limited to hundreds of nanometers in the surface layer.

## Figures and Tables

**Figure 1 materials-15-01851-f001:**
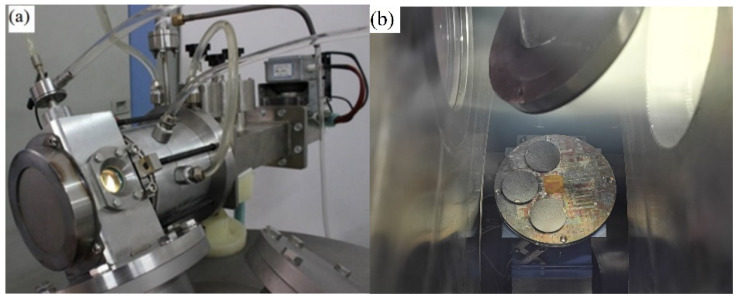
(**a**) AO simulation device, (**b**) picture of samples exposing by AO, (**c**) structure schematic of AO emission device.

**Figure 2 materials-15-01851-f002:**
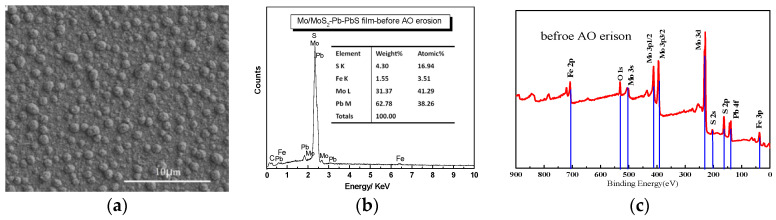
SEM morphology, EDS composition, and XPS survey of Mo/MoS_2_–Pb-PbS composite film before and after AO erosion: (**a**–**c**) without AO erosion. (**d**–**f**) AO erosion.

**Figure 3 materials-15-01851-f003:**
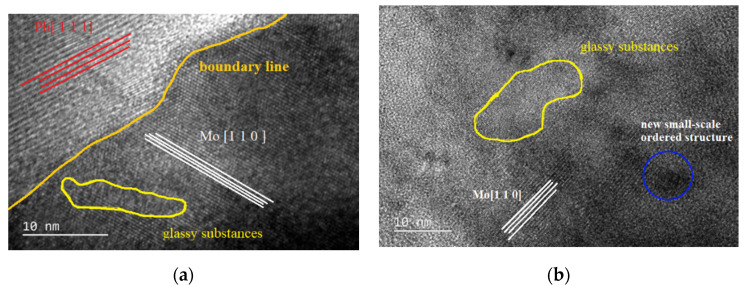
TEM images of Mo/MoS_2_-Pb-PbS composite film before and after AO erosion: (**a**) composite film without AO erosion, (**b**) composite film with AO erosion.

**Figure 4 materials-15-01851-f004:**
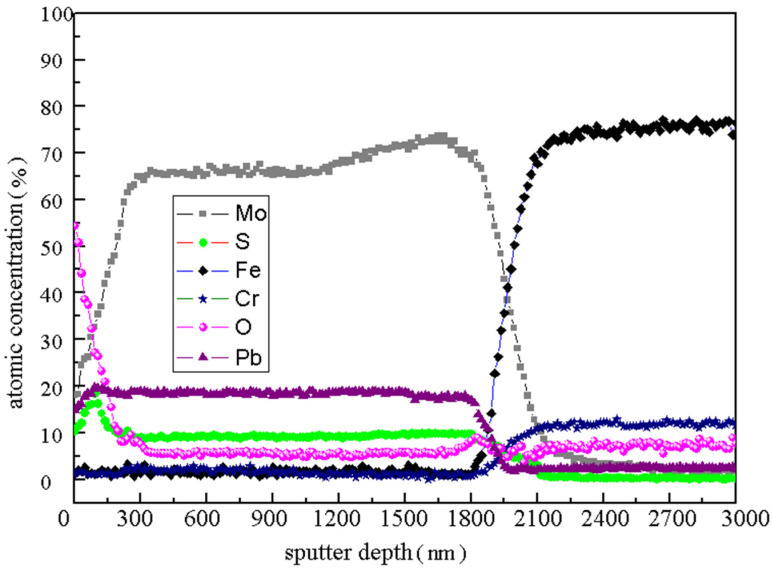
AES Etching Test Results of Mo/MoS_2_-Pb-PbS Film in Depth Direction after AO Erosion.

**Figure 5 materials-15-01851-f005:**
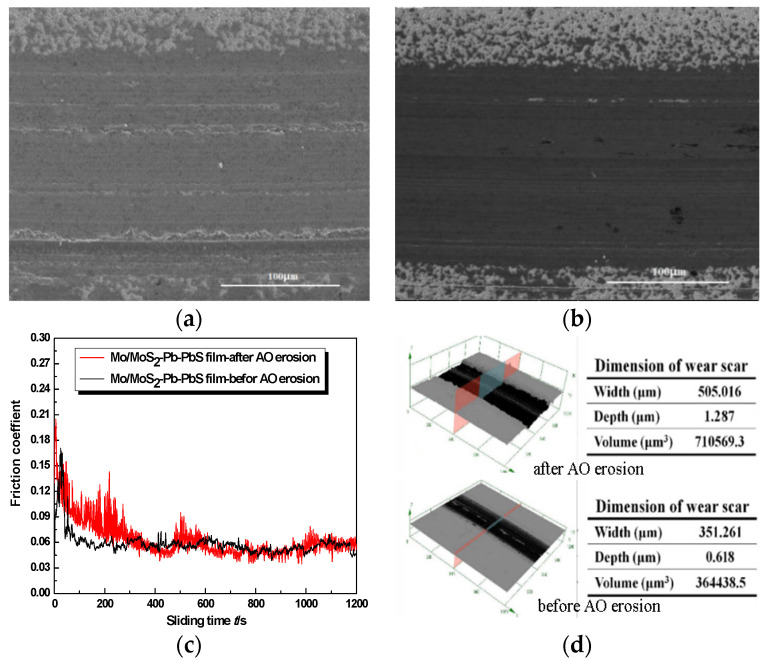
Comparison of tribological properties of Mo/MoS_2_-Pb-PbS composite film before and after AO treatment: (**a**) The wear scar morphology before AO erosion. (**b**) The wear scar morphology after AO erosion. (**c**) Friction coefficient curve. (**d**) 3D wear scar morphology.

**Figure 6 materials-15-01851-f006:**
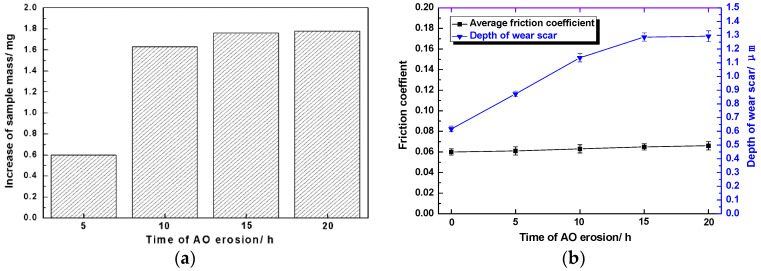
The mass and tribological properties of Mo/MoS_2_-Pb-PbS thin film samples after AO treatment at different times: (**a**) Mass change. (**b**) Average friction coefficient and variation in wear scar depth.

**Figure 7 materials-15-01851-f007:**
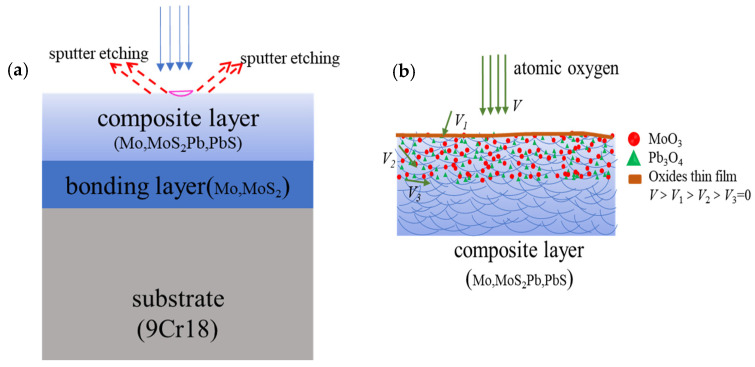
Schematic of AO etching and oxidation: (**a**) Schematic diagram of sputter-etching; (**b**) Diagram of oxidation.

## Data Availability

Not applicable.

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
