# Peer review of "Research on Atomic Oxygen Erosion Influence of Structural Damage and Tribological Properties of Mo/MoS2-Pb-PbS Thin Film"

_materials, 2022, doi:10.3390/ma15051851_

Round 1
Reviewer 1 Report
The manuscript “Research on atomic oxygen erosion influence of structural damage and tribological properties of Mo/MoS2-Pb-PbS thin film” is an interesting piece of work.
- The abstract needs to be more substantial, clearly bringing out the motivation and novelty of the work.
- In fig. 6(a), it seems that the standard deviation of all the four different columns is the same. How and why is this?
- The conclusion needs to be more robust and specific. Provide conclusion in bulleted points.
Author Response
1 The abstract needs to be more substantial, clearly bringing out the motivation and novelty of the work.
Response: Thank you for your kind advices. We have added brief sentence of the motivation in the manuscript with green highlight.
2 In fig. 6(a), it seems that the standard deviation of all the four different columns is the same. How and why is this?
Response: Thank you for your careful work. We are sorry for our careless work. The standard deviation was not analyzed, we had changed the picture.
3 The conclusion needs to be more robust and specific. Provide conclusion in bulleted points.
Response: Thank you for your kind advices. We changed the conclusion used the way of bulleted points. you can find the changes from manuscript
Reviewer 2 Report
The article presents the results of studying the effect of atomic oxygen on the mechanical properties and corrosion resistance of Mo/MoS2-Pb-PbS films. The following methods of SEM, TEM, XPS analysis of the film surface were used for the experiments. The authors have established a number of dependencies that are of great scientific interest, in view of their novelty. This research topic is one of the most relevant today, and the selected research objects have a high potential for practical application. However, after reading the reviewer, a number of questions arose that the authors should answer before the work can be published.
1 In the introduction, authors should draw attention to similar studies and provide a brief overview of the current state of research in this area.
2. The authors should explain the presence of Hillock-like structures on the surface of the films shown in Figure 2.
3. Authors should compare the change in the concentration of hillocks before and after erosion testing.
4. The authors should explain what the formation of glass-like inclusions means and how exactly they are formed in the original samples.
5. Why does the coefficient of friction decrease with the number of cycles?
6. Authors should describe the corrosion mechanism in more detail.
Author Response
Dear reviewer:
Thank you for your kind suggestions about our paper, ‘Research on atomic oxygen erosion influence of structural damage and tribological properties of Mo/MoS2-Pb-PbS thin film’ (materials-1605221). These comments are all valuable and very helpful for revising and improving our paper, as well as the important guiding significance to our researches. We revised the manuscript in accordance with the your comments, and carefully proof-read the manuscript to minimize typographical, grammatical, and bibliographical errors. We have made all changes into a green shading in the revised submission for easy tracking. A list of responses to reviewers' comments was as follows.
If there were any questions, please contact with me.
Sincerely yours,
Cuihong Han
The response of comments
1 In the introduction, authors should draw attention to similar studies and provide a brief overview of the current state of research in this area.
Response: Thank you for your kind advice. We marked the brief overview sentences by green shading, and added some literature.
- The authors should explain the presence of Hillock-like structures on the surface of the films shown in Figure 2.
Response: Thank you for your careful work. Because flash etching of AO beam, the film surface became flat and full of pit-ting. Around pit-ting, the peaks were formed, surface film appeared Hillock-like structures. we explained the etching by graphic and sentence, you can find them in the manuscript marked by green shading.
- Authors should compare the change in the concentration of hillocks before and after erosion testing.
Response: Thank you for your good suggestion. The concentration of hillocks before and after erosion testing related with AO beam density, we are sorry about our aborting. I regrated the we cannot do any test in plan because of the Coronavirus happened in Tianjin China. We will concentrate the research of concentration of hillocks when all goes well. Thank you again.
- The authors should explain what the formation of glass-like inclusions means and how exactly they are formed in the original samples.
Response: Thank you for your careful comments. We are sorry about our loose. the description of ‘glassy substances’ was one crystal structure of 2H which is the stable state structure. Molybdenum disulfide crystal belongs to hexagonal system, with three crystal structures of 1T, 2H and 3R. IT-MoS2 and 3R-MoS2 crystal belonged to metastable structure. The metastable states of 1T-MoS2 and 3R-MoS2 can be transformed into stable 2H-MoS2 at high temperature caused by highly active atomic oxygen impact. Hence, the glassy substance increased after AO erosion.
- Why does the coefficient of friction decrease with the number of cycles?
Response: Thank you for your good question. Because oxide thin layer was removed with the number of cycles, the friction-reducing lubrication performance of the film recovered quickly.
- Authors should describe the corrosion mechanism in more detail.
Response: Thank you for your insightful suggestion. We described the discussion section again, and this section was marked by green shading.

Reviewer 3 Report
The manuscript is well structured but needs major revisions in most of its parts.
1) Once defined atomic oxygen (AO), do not use atomic oxygen but its abbreviation.
2) Do not use an abbreviation or a chemical composition in keywords
3) Pay attention to paragraph titles
4) IMPORTANT: There is not a reason for describing four AO erosion experiments for four erosion times in the process section and, at the end, describe only the 15 hours erosion time. Please, the manuscript should be focused only on the 15 hours experiments explaining this choice. For instance, it is the minimum time to have erosion measurable effects.
5) IMPORTANT: A comparison between before and after AO erosion is necessary for all measurements presented in the manuscript. This is fundamental for understanding physical/chemical effects induced by AO.
6) IMPORTANT: Figure 2 has to be changed. Graphs (b, e) and (c, f) should be combined for comparison purposes.
7) Figure 4 is not clear as well as the manuscript part where Auger measurements are described. Ar sputtering is used to erode the film and this process should be described together with an evaluation of erosion depth. A graph plotting chemical elements vs depth is expected.
8) Tribological properties should be described before and after AO erosion.
9) Mechanism section seems more a Discussion section and needs to be improved.
Author Response
Dear reviewer:
Thank you for your kind suggestions about our paper, ‘Research on atomic oxygen erosion influence of structural damage and tribological properties of Mo/MoS2-Pb-PbS thin film’ (materials-1605221). These comments are all valuable and very helpful for revising and improving our paper, as well as the important guiding significance to our researches. We revised the manuscript in accordance with the your comments, and carefully proof-read the manuscript to minimize typographical, grammatical, and bibliographical errors. We have made all changes into a green shading in the revised submission for easy tracking. A list of responses to reviewers' comments was as follows.
If there were any questions, please contact with me.
Sincerely yours,
Cuihong Han
The response of comments
The response of reviewer 3th
The manuscript is well structured but needs major revisions in most of its parts.
- Once defined atomic oxygen (AO), do not use atomic oxygen but its abbreviation.
Response: Thank you for your kind advice. We are sorry about this and had changed this mistake. You can find them form our manuscript marked by green shading.
- Do not use an abbreviation or a chemical composition in keywords
Response: Thank you for your valuable comments. We are sorry of our careless writing and had changed this mistake and marked them by green shading.
- Pay attention to paragraph titles
Response: Thank you for your careful work. We checked the paper carefully and changed the incorrect sentence. And you can find these changes that are marked with green shading in this paper.
- IMPORTANT: There is not a reason for describing four AO erosion experiments for four erosion times in the process section and, at the end, describe only the 15 hours erosion time. Please, the manuscript should be focused only on the 15 hours experiments explaining this choice. For instance, it is the minimum time to have erosion measurable effects.
Response: Thank you for your valuable advice. Based on your comments, we analyzed the weight changes and the relation of initial friction coefficient and wear scar in 5 h, 10h, 15 h, 20h four periods. The manuscript may be bloated if we analyzed all the SEM, EDS, XPS about four periods, so we choose AO erosion for 15 hours which is representative in morphology and composition. We are sorry about this.
- IMPORTANT: A comparison between before and after AO erosion is necessary for all measurements presented in the manuscript. This is fundamental for understanding physical/chemical effects induced by AO.
Response: Thank you for your insightful advice. According to your advice, we compare the SEM, EDS composition and XPS survey, tribological properties of Mo/MoS2-Pb-PbS composite film. All the changed description was marked with green shading in our manuscript.
6) IMPORTANT: Figure 2 has to be changed. Graphs (b, e) and (c, f) should be combined for comparison purposes.
Response: Thank you for your valuable and insightful advice. Based on the advice received,we explained the graphs(b, e) and (c, f) and compared the change of elements before and after AO erosion. The results showed that the content of MoS2 decreased significantly because a large amount of Mo was oxidized to MoO3 and metastable MoO2 Combining with Figure 2 (e) and (f). An obvious feature in the spectrum is the increase in the content of elemental Mo on the film sur-face caused by AO erosion because sputtering etching effect of high-energy AO removed the S-rich layer on the film surface. The Pb element on the original film surface mainly exists in the form of elemental Pb, PbS and PbO, and the content of PbO was a little. After AO erosion, the Pb element on the film surface was oxidized to Pb3O4 and minor PbSO3, and the spectral peak of elemental Pb disappeared.
7) Figure 4 is not clear as well as the manuscript part where Auger measurements are described. Ar sputtering is used to erode the film and this process should be described together with an evaluation of erosion depth. A graph plotting chemical elements vs depth is expected.
Response: Thank you for your valuable comments. We are sorry that we omitted important information of the speed in fig 4. In the manuscript, we introduced the speed of sputtering and the fig.4 x-axis was sputtering time, the depth was the multiplication of speed and time. Based on your comments, we improved figure 4 and x-axis was the sputtering depth.
8) Tribological properties should be described before and after AO erosion.
Response: Thank you for your kind advice. We supplemented the description of the tribological properties before AO erosion, and the supplement’s part was marked by green shading.
9) Mechanism section seems more a Discussion section and needs to be improved.
Response: Thank you for your good suggestion. We described the discussion section again, and this section was marked by green shading.

Round 2
Reviewer 1 Report
The authors have responded to my comments and thus, I recommend its acceptance.
Reviewer 2 Report
The authors answered all the questions, the article can be accepted for publication.
Reviewer 3 Report
Requirements are mostly fulfilled and now the manuscript is more clear